# Portuguese Neonatal Screening Program: A Cohort Study of 18 Years Using MS/MS

**DOI:** 10.3390/ijns10010025

**Published:** 2024-03-20

**Authors:** Maria Miguel Gonçalves, Ana Marcão, Carmen Sousa, Célia Nogueira, Helena Fonseca, Hugo Rocha, Laura Vilarinho

**Affiliations:** 1Department of Human Genetics, National Institute of Health Doutor Ricardo Jorge, 4000-053 Porto, Portugal; ana.marcao@insa.min-saude.pt (A.M.); carmen.sousa@insa.min-saude.pt (C.S.); celia.nogueira@insa.min-saude.pt (C.N.); helena.fonseca@insa.min-saude.pt (H.F.); hugo.rocha@insa.min-saude.pt (H.R.); 2Faculty of Sciences, University of Lisbon, 1749-016 Lisboa, Portugal

**Keywords:** Portuguese neonatal screening program, neonatal screening, inborn errors of metabolism (IEM), second-tier testing (2TT)

## Abstract

The Portuguese Neonatal Screening Program (PNSP) conducts nationwide screening for rare diseases, covering nearly 100% of neonates and screening for 28 disorders, including 24 inborn errors of metabolism (IEMs). The study’s purpose is to assess the epidemiology of the screened metabolic diseases and to evaluate the impact of second-tier testing (2TT) within the PNSP. From 2004 to 2022, 1,764,830 neonates underwent screening using tandem mass spectrometry (MS/MS) to analyze amino acids and acylcarnitines in dried blood spot samples. 2TT was applied when necessary. Neonates with profiles indicating an IEM were reported to a reference treatment center, and subsequent biochemical and molecular studies were conducted for diagnostic confirmation. Among the screened neonates, 677 patients of IEM were identified, yielding an estimated birth prevalence of 1:2607 neonates. The introduction of 2TT significantly reduced false positives for various disorders, and 59 maternal cases were also detected. This study underscores the transformative role of MS/MS in neonatal screening, emphasizing the positive impact of 2TT in enhancing sensitivity, specificity, and positive predictive value. Our data highlight the efficiency and robustness of neonatal screening for IEM in Portugal, contributing to early and life-changing diagnoses.

## 1. Introduction

The Portuguese Neonatal Screening Program (PNSP) began on 14 May 1979 with the screening for phenylketonuria (PKU). Over the years, the panel of screened conditions was expanded to the current 28 conditions and has allowed for the diagnosis of more than 2400 affected neonates. The panel includes 24 inborn errors of metabolism (IEMs), congenital hypothyroidism, cystic fibrosis, sickle cell disease, and spinal muscle atrophy (ongoing pilot study). Despite the non-mandatory nature of this program, its coverage is approximately 100% of neonates born in Portugal, with around 350 samples processed daily in the Neonatal Screening Unit (NSU) at the National Institute of Health Doutor Ricardo Jorge [1].

In 2004, the NSU implemented a tandem mass spectrometry (MS/MS) pilot study to test for acylcarnitines and amino acids [1]. This upgrade allowed for the simultaneous screening of multiple conditions in a single sample, increasing the analytic ability of the PNSP and prompting a nationwide expanded neonatal screening (NBS) program for IEM, officially established in 2006. The panel included 23 other metabolic disorders in addition to PKU.

IEMs are rare genetic disorders that cause disruption of metabolic pathways due to an enzyme cofactor or transporting protein defect. The metabolic disruption can cause intoxication-type disorders or energy metabolism disorders. Toxic accumulation disorders are due to a build-up of metabolites before the blockage site and/or to the deficiency of a product, with possible diversion of accumulated substrates to alternative pathways, originating very toxic metabolites. Energy metabolism disorders are due to defects in energy production or utilization [2]. Individually rare but collectively common, these disorders benefit from early identification. Early treatment can avoid symptom onset, preventing sequalae and improving the overall prognosis [3]. IEMs screened with the PNSP are categorized into four groups: amino acid disorders (AAD), urea cycle disorders (UCD), organic acidemias (OA), and mitochondrial fatty acid β-oxidation disorders (FAOD). The main goal of IEM-NBS is first to identify suspected biochemical patients and forward them to multidisciplinary teams to potentially confirm the diagnosis and initiate clinical treatment.

The initial results of the pilot study started in 2004 were published in 2010 [1]. Throughout the years, second-tier testing (2TT) was introduced in expanded NBS to increase the sensitivity and specificity of screening for disorders with higher number of false positives (FP) or false negatives (FN).

Now, after 18 years of NBS by MS/MS, the data obtained from 2004 to 2022, as well as the overall evolution of expanded NBS, are presented in this work.

## 2. Materials and Methods

### 2.1. Studied Subjects

From 2004 to 2022, 1,764,830 neonates were screened with MS/MS technology. Dried blood spot (DBS) samples (Guthrie cards) from neonates nationwide were collected in public and private hospitals and primary health centers, and then sent to the NSU by regular mail (Figure 1). It was recommended that the samples be collected between the 3rd and 6th days of life, with at least 48 h of feeding, as these factors affect the outcome of NBS.

During the first two years of the pilot study, expanded NBS was restricted to the north and central regions of Portugal and then progressively extended to the whole country. From 2006 onward, all Portuguese territory was included. All neonates that displayed suspicious biochemical alterations were referred to Reference Treatment Centers (RTC) for IEMs for clinical follow-up. After an initial clinical evaluation, the patients were further studied to confirm the diagnosis and start treatment. 

### 2.2. Routine Biochemical Screening

Routine IEM-NBS was performed through MS/MS [4], initially using two API 2000 spectrometers (Applied Biosystems, Waltham, MA, USA), and later one 4000 QTRAP and one 4500 Triple Quad LC-MS/MS (Sciex, Framingham, MA, USA).

A single 3.2 mm DBS was placed in a 96-well, V-shaped polypropylene plate. Methanol-diluted internal standards for amino acids and acylcarnitines (NSKAB mixture from Cambridge Isotope Laboratories, Tewksbury, MA, USA) were added (100 µL of a 1:100 dilution). After gently shaking for 25 min, the extract was transferred to a flat-bottom plate and hot-dried under a gentle nitrogen flow. Butanol-HCl (60 µL) was then added, and the covered plate was incubated at 70 °C for 15 min. The dried extract was finally reconstituted with acetonitrile and water (50:50). After covering with aluminum foil, the samples were analyzed through MS/MS.

Biochemical screening criteria, as well as primary and secondary biomarkers, cut-offs, and ratios, were adjusted over the years in order to increase specificity and sensitivity, and the ones presently in use are specified in Table 1.

### 2.3. Second Tier Tests (2TT)

The NSU currently uses 2TT for succinylacetone (SA), methylmalonic acid (MMA), total homocysteine (tHcy), 3-OH-propionic acid (3OHprop), propionyl-glycine (PropGly), and isovalerylcarnitine/2-methylbutyrylcarnitine (C5)/pyvanoylcarnitine (Piv-C5) determinations (Table 2 and Table 3).

Tyrosine is the primary biomarker for the detection of tyrosinemia. However, this marker’s elevation has low specificity since hypertyrosinemia in neonates can be associated with different factors, such as transient hypertyrosinemia of the neonate, liver disease, or congenital enzymatic deficiencies [9]. The choice of tyrosine as a primary marker for TYR 1 is justified by the age at sample collection. In Portugal, samples are collected between 3 and 6 days, which allows for an increase in tyrosine.

To increase the sensitivity and specificity of the screening for tyrosinemia type I (TYR1), the PNSP implemented the determination of SA, which is a pathognomonic biomarker for TYR1. The distinction between tyrosinemia type II and III is only possible after genetic study.

Propionylcarnitine (C3) is a primary biomarker for the detection of PA, MMA, and cobalamin defects. However, C3 has low diagnostic sensitivity, contributing to a high number of FP [7]. This fact led to the implementation of MMA, 3-OHprop, and PropGly determination in DBS as a way to improve the positive predictive value (PPV) of these screenings [6].

Methionine is the primary biomarker for screening of classic homocystinuria (CBS deficiency). However, FP due to secondary methionine elevations can arise from liver disease, parenteral nutrition, or MAT I/III deficiency. This indicates that hypermethioninemia can indicate conditions other than CBS deficiency [10]. The implementation of tHcy determination in DBS increased the sensitivity and specificity of the screening for CBS deficiency.

The screening of isovaleric aciduria (IVA) quantifies isovaleryl/2-methylbutyrylcarnitine (C5), a marker with a low positive predictive value. This fact motivated the implementation of 2TT for isovalerylcarnitine, 2-methylbutyrylcarnitine, and pyvaloylcarnitne. Since elevations of C5-acylcarnitines in DBS can indicate IVA or 2-methylbutyryl-CoA dehydrogenase deficiency, it is important to have a method for differential diagnosis that is faster than conventional urinary acid and acylglycine analysis [8].

## 3. Results and Discussion

Over the last 18 years, a total of 1,764,830 neonates were screened in Portugal for an extended group of 24 metabolic disorders, and a total of 677 patients were detected (Table 4), with an overall birth prevalence of 1:2607.

Of the screened IEMs, the one with the highest birth prevalence in Portugal is MCAD deficiency (1:6603). This is one of the highest prevalences reported [16], which is due to the fact that the great majority of the patients (more than 95%) were from Gypsy communities, which are known to have high inbreeding rates with a high prevalence of genetic disorders [17]. A total of 87% of Portuguese patients presented the most frequent mutation, the c.985A>G (p.Lys329Glu) mutation in the ACADM gene, in homozygosity. Among the organic acidurias, two can be emphasized: 3-MCCD, which is the most frequent and has already been reported [18], and CblC deficiency, which is found in our country with one of the highest birth prevalences worldwide. Genotype–phenotype correlations of Portuguese CblC and CblD patients have already been discussed further [19]. The most common MMACHC mutation was c.271dupA, present in 100% of the MMACHC alleles of all CblC-screened patients [19].

Among the detected disorders were not only those included as targets of the Portuguese NBS program (Table 1), but also differential diagnoses, namely, holocarboxylase synthase deficiency (HLCS deficiency), β-ketothiolase deficiency (BKTD), Brown–Vialetto–Van Laere Syndrome (BVVL), mitochondrial trifunctional protein deficiency (MTPD), and carnitine-acylcarnitine translocase (CACT) deficiency.

One case of BKTD was identified with an increase in C5-OH (3-hydroxy-isovalerylcarnitine/2-methyl-3-hydroxy-butyrylcarnitine) (1.51 µM, cut-off < 0.3 µM) alongside with C5:1 (tiglylcarnitine) (0.28 µM, cut-off < 0.07 µM). Molecular analysis confirmed the BKTD diagnosis, and with the identification, the pathogenic mutations c.455G>C (p.Gly152Ala) [20] and c.814C>T (p.Gln272*) [21] were found in the ACAT1 gene. The vast majority of BKTD-affected individuals display metabolic decompensation and improve with early treatment and management [22,23], making this condition eligible to be included in our NBS panel in the future, since it is compliant with the Wilson and Jungner criteria. Also associated with an elevated value of C5-OH (being a differential diagnosis of 3-MCCD), two cases of HLCS deficiency were found. One neonate had a c.2127G>T(p.Pro708Pro) polymorphism in heterozygosity, whose diagnosis was confirmed via OA analysis, and one case had the novel mutation c.995A>G (p.Gln332Arg) in homozygosity in the HLCS gene.

One case of Brown–Vialetto–Van Laere Syndrome (BVVL) was detected, with a MADD-like acylcarnitine profile. The neonate initially presented elevated C4 and C5DC, and a second sample analysis revealed a MADD-like suggestive acylcarnitine profile (C4, C5DC, C6, C8, C10 and C10:1 elevations). After the exclusion of ETFA/ETFB/ETFDH mutations, the c.383C>T (p.Ser128Lys) mutation, which has previously been reported, was found in the SLC52A2 gene in homozygosity [24].

One case of MTPD was detected as a differential diagnosis of LCHADD, from which it is completely indistinguishable based on the acylcarnitine profile. This patient presented the mutation c.1137delT (p.His379Glnfs*76), which has previously been reported [25], and the novel mutation c.1062-?_c.1425+?del in the HADHB gene.

Similarly, two cases of carnitine-acylcarnitine translocase (CACT) deficiency were detected as differential diagnoses of carnitine palmitoyl-transferase II deficiency (CPTII). These patients presented two distinct genotypes: one was homozygous to the c.84delT (p.His29Thrfs*100) mutation [26] and the other was homozygous for the c.326+1delG mutation in the SLC25A20 gene [27].

Although not included in the panel of screened disorders, in all these situations, timely intervention was possible in the neonates.

Maternal disorders/conditions, situations in which the maternal condition is responsible for the observed biochemical phenotype of the neonate, were also detected through the NBS of their children. Mothers were detected with MCADD, GA1, CUD, 3-MCC, and maternal B12 deficiency (Table 5). In the cases of neonates from mothers with MCADD, GA1, and CUD, the major observed abnormality was a severe decrease in free carnitine. In the neonates from mothers with 3-MCCD, increases in C5OH and/or decreases in free carnitine were observed. And in neonates of mothers with vitamin B12 deficiency, increases in C3 and C3/C2, with MMA and/or tHcy in the 2TT, were the pattern. The identification of these situations allowed for prompt intervention in the neonates, namely, carnitine supplementation or correction of vitamin B12 status, with a positive impact on the clinical outcomes of the neonates [28]. It also allowed for a clinical evaluation of the affected mothers, improving their clinical statuses and decreasing the risk of potential decompensation episodes [29]. Although there are growing reports of uneventful pregnancies in IEM patients, knowledge of an IEM can reduce complications and help with future pregnancies. It is evident that the discovery of maternal IEM is one of the long-term benefits of NBS [29].

NBS programs should be dynamic and adjust themselves to maximize their impact in public health. Since 2004, several adjustments have been made to the screening criteria of several metabolic disorders, including cut-off adjustments, adoption of new screening markers, and introduction of 2TT. One of the alterations made was related to the screening of methionine adenosyltransferase deficiency, which was initially screened based on the elevation of methionine over 65 μM. Using this approach, several neonates with MAT I/III were identified; 95% of these patients were heterozygous for c.791G>A (p.Arg264His) mutation in MAT1A [10,30], and clinical follow-up revealed that most of them rarely presented significantly elevated methionine levels and remained asymptomatic. Based on this observation, the Portuguese NBS program advisory board suggested sending for clinical evaluation only those neonates that presented with methionine over 300 μM on a confirmation sample. This strategy has been followed since 2019, and most patients with heterozygosity for the c.791G>A (p.Arg264His) dominant mutation are probably no longer identified.

Another example is the detection of arginase deficiency. Since implementing the use of the ratio Arg/Orn as the main marker for the screening of this disease, the number of FP cases has dramatically decreased, as hyperargininemia resulting from secondary causes (i.e., dietary intake, such as parenteral nutrition [31]) could be excluded.

Since 2017, an important addition to the biochemical screening of IEM has been the introduction of 2TT, performed on a screening sample when a marker for a disorder is elevated. This additional step is used to increase the specificity and sensitivity of NBS. Overall, 2TT increases the efficiency of screening programs and minimizes FP and false FN results [32].

An example of the influence of 2TT on screening efficiency is IVA NBS. From 2011 to 2015, an unusually high number of false positives (FP) for IVA were detected. These cases presented elevated C5, which can be an indication of IVA. This atypical increase in this biomarker was discovered to be due to the presence of neopentanoate, a substance used as an emollient in the cosmetics industry [33]. After surveying mothers, it was concluded that the FP results were caused by the use of a nipple fissure cream used by lactating mothers. These findings were also reported by other European neonatal screening programs [34]. After the implementation of 2TT for C5 isomers, the number of FPs for IVA decreased to zero (Table 6), which makes the impact of the secondary biomarker clear. Overall, the implementation of 2TT increased the positive prediction of IEM-NBS values from 21 to 31% (2017) and decreased the FP rate from 0.17% (2016) to 0.10% (2017). Another advantage of 2TT is the increase in sensitivity for some disorders, namely, those that use C3 as a primary marker (by allowing for a decrease in the C3 cut-off).

The overall recall rate decreased, with a pronounced effect, after 2017 due to 2TT implementation (Figure 2). In our experience, besides the aforementioned FP cases of C5 increases, the biomarkers most associated with sample recall are C5DC, C3, and arginine.

The information extracted from 18 years of MS/MS screening is extensive and requires a detailed analysis. The parameters of extended NBS are presented in Table 7. The average age at referral for a positive test in the MS/MS era is 10.1 days (Figure 3), which displays the efficiency of the PNSP.

FP can be the result of multiple factors that should be considered by NBS programs. It is estimated that between 10 and 40% of FPs are due to low birth weight, prematurity, neonatal sickness, total parenteral nutrition feeding (TPN), or medications [35]. Concerning prematurity, it is known that preterm neonates can present enzymatic immaturity, displaying multiple alterations in NBS. Second sample requests, at a later time, can help to address some of these issues and prevent the reporting of these cases to RTC.

There have also been reports of FP cases associated with ethnic variability in NBS markers for some IEMs. Evidence of an association between ethnicity status and the levels of NBS markers for GA1, MMA, and VLCADD has been presented, suggesting that this can be a cause of FP for the screening of these disorders [36]. Migrations, refugee crises, and other population-shifting events may require future changes in the PNSP. In 2022, 14,003 out of the 83,671 live births in Portugal were babies born to foreign mothers. This means that at least 16.7% of neonates born in 2022 had foreign ancestry [37]. Therefore, considering the significant contribution of immigrants to the Portuguese birth rate, ethnic variability of NBS biomarkers may be a factor to explore further.

We classified as FNs cases with normal NBS with posterior positive diagnosis of a PNSP disorder. Eight cases were identified which fit the FN criteria. Two siblings with argininosuccinic aciduria (or ASL deficiency) had the late-onset form of this disease. They presented clinical symptoms at 4 months of age. Both presented no abnormalities at the time of NBS (normal citrulline and argininosuccinic acid) and displayed the c.35G>A (p.Arg12Gln) mutation in homozygosity in the ASL gene [38].

Four late-onset cases of CPTII (muscular forms) with normal screening were also identified. All patients presented the c.338C>T (p.Ser113Leu) mutation in the CPT2 gene [39], which is known to be associated with the CPTII muscular form. Lastly, two FN cases of MADD (type III) were identified, both homozygous for the novel c.461C>T (p.Tyr154Met) mutation in ETFB, with normal NBS results. It is recognized that the detection of late-onset forms through NBS may be difficult, but attending to the fact that these patients only present symptoms later in life, they are not the primary targets of NBS screening programs [35].

## 4. Conclusions

This study shows the wide range of IEMs that can be identified with MS/MS. Most of the diseases mentioned in this work are rare individually but very significant as a group, making them important to screen.

We highlight the maintenance of around 100% coverage of the neonate population, something that has remained constant over the years and continues in the tandem mass era. The performance metrics of the PNSP are satisfactory, with high sensitivity and a good referral time.

There is potential for the inclusion of additional diseases in the PNSP’s panel. The disorders which can feasibly be detected with our current methodology and that exist in our population can be included in the Portuguese IEM panel. Therefore, BKTD, HLCS deficiency, CACT deficiency, and MTP deficiency can be included in MS/MS for NBS in the future.

This work also shows the way the PNSP has grown and adapted since the publication of the data from the pilot study of MS/MS-expanded NBS. The rare nature of these pathologies affects the obtention of data from a significant number of individuals. With the great volume of epidemiological data acquisition accomplished by this work, we can characterize our cohort of IEM patients. This reflects 18 years of a national effort to identify and treat individuals with rare and otherwise potentially fatal disorders as early as possible, fulfilling the PNSP’s mission.

Since the conclusion of the pilot study for expanded NBS in 2006 [1], technological and scientific advances have shaped the course of PNSP. The discovery of new biomarkers, methodologies, and knowledge of the natural history of IEMs will drive change and further develop NBS.

In the European panorama, much can be done to accomplish a uniformized and efficient network of neonatal screening for rare diseases [40]. However, there is a clear trend of improved collaboration in Europe, which will lead to a consensus and will motivate the expansion of screening programs [41].

In a 2021 study involving 50 European countries [41], Portugal was found to be one of 19 that screens for 20 or more disorders with MS/MS. Moreover, the PNSP follows the centralized laboratory approach for maximized efficiency, which is adequate for the number of infants screened per year and remains the most effective and economically advantageous strategy for NBS in Portugal [41].

The approximately 100% coverage of the PNSP is due in part to the centralization of screening in one laboratory, which allows for more control over the standardization of the screening methodology, also contributing to an easier analysis of the data. This yields more reliable and timely results, helping hundreds of families with both answers and early treatment, which provides better prognoses, life quality, and outcomes. On the other hand, for the past 42 years, NBS has been an integral part of neonatal health, with parents and families organically participating in the PNSP, which demonstrates its stability and good reputation.

## Figures and Tables

**Figure 1 IJNS-10-00025-f001:**
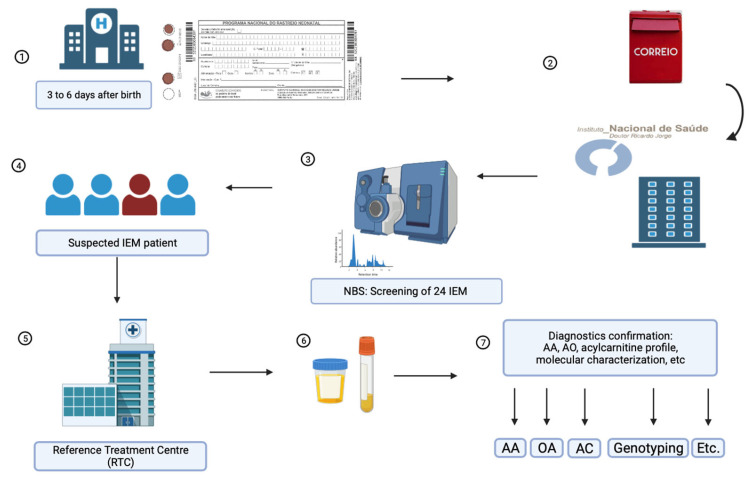
Schematic representation of sample processing for IEM-NBS by the PNSP. **1.** DBS samples from all Portuguese territory were collected at public and private hospitals and primary health centers. **2.** Samples were forwarded to the NSU at the National Institute of Health Doutor Ricardo Jorge. **3.** All samples are tested for amino acids and acylcarnitines as butyl esters, and subjected to 2TT using MS/MS. **4.** Suspected IEM individuals were forwarded to an RTC. **5.** RTC performed a clinical evaluation and collected additional patient samples. **6.** Urine and blood samples were sent to the NSU for diagnostic confirmation. **7.** Diagnosis confirmation included amino acids (AA), urinary organic acids (OA), acylcarnitines (AC), molecular characterization (genotyping), and other tests, when appropriate.

**Figure 2 IJNS-10-00025-f002:**
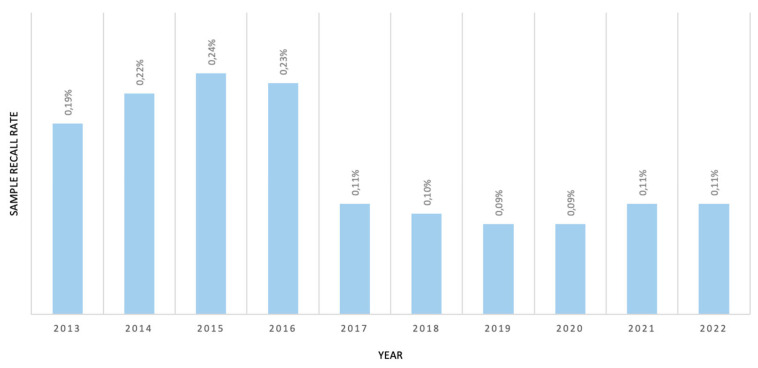
Sample recall rate due to IEM biomarker alterations from 2013–2022.

**Figure 3 IJNS-10-00025-f003:**
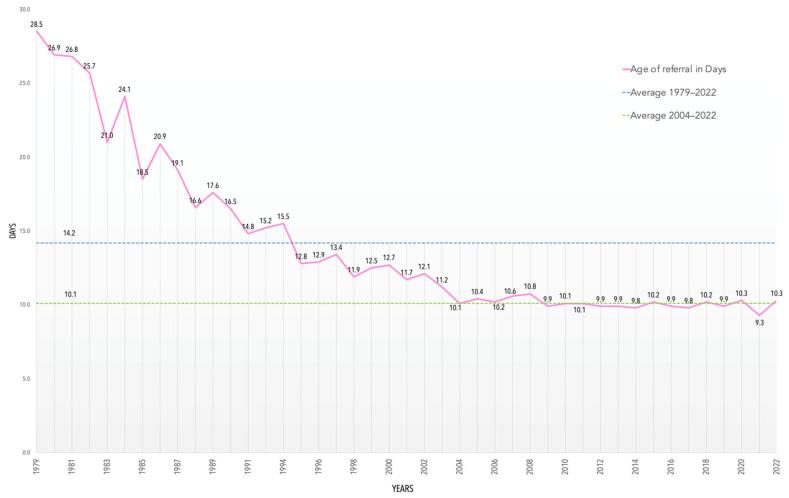
Average age at referral in days from 1979 to 2022.

**Table 1 IJNS-10-00025-t001:** Screening criteria for disorders detected by expanded neonatal screening with MS/MS.

Screened Disorders		OMIM	Cut-Offs and Ratios
**Amino Acid Disorders (AAD)**	Phenylketonuria (PKU)/Hyperphenylalaninemia (HPhe)	261600	Phe (>150 μM) and Phe/Tyr (>1.5)
	Maple syrup urine disease (MSUD)	248600	XLeu (>270 μM) and Val (>285 μM) Val/Phe > 4, Xleu/Phe > 5
	Tyrosinemia type I (TYR1)	276700	Tyr (>210 μM) selects for 2TT (SA)
	Tyrosinemia type II (TYR2)	276600	Tyr (>500 μM) selects for 2TT
	Tyrosinemia type III (TYR3)	276710	Tyr (>500 μM) selects for 2TT
	Homocystinuria (CBS deficiency)	236200	Met (>45 μM) selects for 2TT
	Methionine adenosyltransferase deficiency (MATI/III deficiency)	250850	Met (>45 μM) selects for 2TT
**Organic Acidurias (OA)**	3-Methyl crotonyl-CoA carboxylase deficiency (3-MCCD)	210200	C5OH (>1.0 μM)
	Isovaleric acidemia (IVA)	243500	C5 (>1.0 μM) selects for 2TT
	Propionic acidemia (PA)	606054	C3 (>5.25 μM) or C3/C2 (>0.2) selects for 2TT
	Methylmalonic acidemia (MMA Mut-/Mut0)	251000	C3 (>5.25 μM) or C3/C2 (>0.2) selects for 2TT
	Malonic acidemia (MAL)	248360	C3DC (>0.35 μM)
	Glutaric acidemia type I (GA1)	231670	C5DC (>0.2 μM)
	3-Hydroxy-3-methylglutaryl CoA lyase deficiency (3HMGLD)	246450	C5OH (>1.0 μM) and C6DC (>0.07 μM)
	Methylmalonic acidemia type CblA/B (MAHCA or CblA and MAHCB or CblB)	251100 251110	C3 (>5.25 μM) and C3/C2 (>0.2) selects for 2TT
	Methylmalonic acidemia type CblC/D (MAHCC or CblC and MAHCD or CblD)	277400 277410	C3 (>5.25 μM) or C3/C2 (>0.2) selects for 2TT
**Urea Cycle Disorders** **(UCD)**	Citrullinaemia type I (CTLN1)	215700	Cit (>200 μM)
	Argininosuccinate lyase deficiency (ASL deficiency)	207900	ASA (>1 μM)
	Arginase deficiency (ARG deficiency)	207800	Arg (>50 μM) and Arg/Orn (>1.0)
**Fatty Acid Oxidation Disorders** **(FAOD)**	Medium-chain acyl-CoA dehydrogenase deficiency (MCADD)	201450	C8 (>0.3 μM) and C8/C10 (>2.5)
	Long-chain 3-OH acyl-CoA dehydrogenase deficiency (LCHADD)/Trifunctional Protein deficiency (TFP)	609016 609015	C16OH (>0.10 μM), C18:1OH (>0.07 μM), C18OH (>0.06 μM) and C16OH/C16 (>0.04)
	Multiple acyl-CoA dehydrogenase deficiency (MADD)	231680	Multiple elevations from C4 to C18 acyl carnitines
	Carnitine uptake defect (CUD)	212140	C0 (<6.8 μM)
	Very-long-chain acyl-CoA dehydrogenase deficiency (VLCADD)	201475	C14:1 (>0.46 μM), C14:2 (>0.17 μM) and C14:1/C12:1 (>6.0)
	Short-chain 3-hydroxyacyl-CoA dehydrogenase deficiency (SCHADD)	231530	C4OH (>0.95 μM)
	Carnitine palmitoyl-transferase I deficiency (CPTIA)	255120	C0/(C16 + C18) (>30)
	Carnitine palmitoyl-transferase II deficiency (CPTII)/Carnitine-acylcarnitine translocase deficiency (CACT)	255110 212138	C0/(C16 + C18) (<3.0)

**Abbreviations:** Phe—phenylalanine; Tyr—tyrosine; Xleu—leucine/isoleucine/allo-isoleucine; Val—valine; CoA—coenzyme A; Met—methionine; C0—free carnitine; C2—acetylcarnitine; C3—propionylcarnitine; C3DC—malonylcarnitine; C4—butyrylcarnitine; C4OH—3-hydroxybutyrylcarnitine; C5—isovalerylcarnitine/2-methylbutyrylcarnitine; C5OH—3-hydroxy-isovalerylcarnitine/2-methyl-3-hydroxy-butyrylcarnitine; C8—octanoylcarnitine; C10—decanoylcarnitine; C5DC—glutarylcarnitine/3-hydroxydecanoylcarnitine; C14:1—tetradecenoylcarnitine; C16—palmityolcarnitine; C16OH—3-hydroxy-palmitoylcarnitine; C18—stearoylcarnitine; C18:1OH—3-hydroxy-oleylcarnitine; C18OH—3-hydroxy-stearoylcarnitine; C6DC—adipoyl/methylglutarylcarnitine; MMA—methylmalonic acid; PropGly—propionylglicine.

**Table 2 IJNS-10-00025-t002:** 2TT markers implemented in the PNSP.

Disorder	Primary Marker	Secondary Marker	Year of Implementation in the PNSP	References
Tyrosinemia	Tyrosine	SA	2006	[5]
Propionic/methylmalonic acidurias	Propionylcarnitine (C3)	MMA, 3OHprop, PropGly	2017	[6,7]
Cobalamin metabolism defects	Propionylcarnitine (C3) and ↓ methionine	MMA and tHcy	2017	[6,7]
Classic homocystinuria	Methionine	tHcy	2017	[7]
Isovaleric aciduria	Isovaleryl/2-methylbutyrylcarnitine (C5)	C5 and Piv-C5	2017	[8]

↓-decreased.

**Table 3 IJNS-10-00025-t003:** Interpretation of IEM-NBS and 2TT results (adapted from [7]).

Condition	2TT Results						
	MMA	PropGly	3OHprop	tHcy	Isovalerylcarnitine	2-Methylbutyrylcarnitine	SA
Propionic acidemia (PA)	N	↑↑	↑↑	N			
Methylmalonyl- CoA mutase deficiency (Mut0 /Mut-)	↑↑	N	N	N			
Cobalamin type A/B deficiency (CblA/B)	↑	N	N	N			
Cobalamin type C/D deficiency (CblC/D)	↑	N	N	↑			
Vitamin B12 deficiency (of maternal cause) ^a^	N or ↑	N	N	N or ↑			
Homocystinuria				↑↑			
Methionine adenosyltransferase I/III deficiency				N or ↑			
Isovaleric aciduria					↑↑		
2-Methylbutyrylglycinuria ^b^						↑	
Tyrosinemia type I							↑↑

^a^ Differential diagnosis of cobalamin metabolism defects. ^b^ Differential diagnosis of Isovaleric acidemia. Not evaluated due to primary markers’ results. N—normal; ↑—increased; ↑↑—highly increased. Abbreviations: MMA—methylmalonic acid; PropGly—propionylglycine; 3OHprop—3-Hydroxypropionic; tHcy—total homocysteine; SA—succinylacetone.

**Table 4 IJNS-10-00025-t004:** Positive cases detected from 2004 to 2022 via NBS screening with MS/MS and birth prevalence in Portugal and worldwide (literature review).

Detected Disorders	Positive Cases	Birth Prevalence	Estimated Worldwide Birth Prevalence
**Amino acid disorders**	**231**	**1:7640**	**1:6803 ^a^**
Phenylketonuria (PKU)/Hyperphenylalaninemia (HPhe)	154	1:11,611	1:15,267 ^a^
Maple syrup urine disease (MSUD)	19	1:92,886	1:81,967 ^a^
Tyrosinemia type I (TYR1)	6	1:294,138	1:100,000 ^a^
Tyrosinemia type II (TYR2)	2	1:882,415	<1:1,000,000 ^b^
Tyrosinemia type III (TYR3)	5	1:294,138	<1:1,000,000 ^b^
Homocystinuria (CBS deficiency)	4	1:441,208	1:243,902 ^a^
Methionine adenosyltransferase deficiency (MATI/III deficiency)	41	*	<1:1,000,000 ^b^
**Urea cycle disorders**	**26**	**1:67,878**	**1:34,364 ^a^**
Citrullinemia type I (CTLN1)	10	1:176,483	1:250,000 ^c^
Argininosuccinate lyase deficiency (ASL deficiency)	9	1:196,092	1:220,000 ^c^
Arginase deficiency (ARG deficiency)	7	1:252,119	1:35,700 ^d^
**Organic acid disorders**	**116**	**1:15,214**	**1:11,481 ^a^**
3-Methyl crotonyl-CoA carboxylase deficiency (3-MCCD)	39	1:45,252	Unknown ^b^
Isovaleric acidemia (IVA)	6	1:294,138	1:196,078 ^a^
Propionic acidemia (PA)	4	1:441,208	1:93,457 ^a^
Methylmalonic acidemia (MMA Mut-/Mut 0)	9	1:196,092	<1:100,000 ^e^
Cobalamin metabolism deficiency (CblA, B, C, and D)/Vitamin B12 deficiency	22	1:80,220	<1:100,000 ^b^
Glutaric acidemia type 1 (GA 1)	20	1:88,219	1:100,000 ^b^
3-Hydroxy-3-methylglutaryl CoA lyase deficiency (3HMGLD)	11	1:160,439	<1:1,000,000 ^b^
ß-Ketothiolase deficiency (BKTD)	1	**	Approximately 250 cases reported worldwide ^f^
Holocarboxylase synthase deficiency (HLCS deficiency)	2	**	1:200,000 ^b^
Malonic acidemia (MAL)	2	1:882,415	<1:1,000,000 ^b^
**Fatty acid oxidation disorders**	**304**	**1:4819**	**1:15,360 ^a^**
Medium-chain acyl-CoA dehydrogenase deficiency (MCADD)	240	1:6603	1:17,301 ^a^
Long-chain 3-OH acyl-CoA dehydrogenase deficiency (LCHADD)	16	1:110,302	1:250,000 ^b^
Mitochondrial trifunctional protein deficiency (MTPD)	1	1:1,764,830	Less than 100 cases reported worldwide ^b^
Short-chain 3-hydroxyacyl-CoA dehydrogenase Deficiency (SCHADD)	2	1:882,415	<1:1,000,000 ^b^
Multiple acyl-CoA dehydrogenase deficiency (MADD)	12	1:147,069	1:200,000 ^b^
Brown–Vialetto–Van Laere Syndrome (BVVL)	1	**	<1:1,000,000 ^b^
Carnitine uptake defect (CUD)	11	1:160,439	Unknown ^b^
Very-long-chain acyl-CoA dehydrogenase deficiency (VLCADD)	12	1:147,069	Over 400 cases reported worldwide ^b^
Carnitine palmitoyl-transferase I (CPTIA) deficiency	4	1:441,208	<1:1,000,000 ^b^
Carnitine palmitoyl-transferase II (CPTII) deficiency	3	1:588,277	<1:1,000,000 ^b^
Carnitine-acylcarnitine translocase (CACT) deficiency	2	1:882,415	Approximately 60 cases reported worldwide ^b^
**Total**	**677**	**1:2607**	**1:1964 ^a^**

* Not calculated due to changes in the screening protocol. ** Not calculated due to not being included in the PNSP panel; ^a^ [11], ^b^ data obtained from Orphanet, ^c^ [12], ^d^ [13], ^e^ [14], ^f^ [15].

**Table 5 IJNS-10-00025-t005:** Cases of maternal disease detected by the PNSP.

Maternal Condition/Disorder	Number of Cases
Vitamin B12 deficiency	27
3-MCCD	18
CUD	8
GA 1	5
MCADD	1
**Total**	**59**

**Table 6 IJNS-10-00025-t006:** Number of FPs for IVA from 2011 to 2017.

Year	Number of FPs for IVA
2011	29
2012	33
2013	59
2014	35
2015	36
2016 ^1^	5
2017 ^2^	0

^1^ Beginning of implementation of 2TT for IVA; ^2^ after implementation of 2TT for IVA.

**Table 7 IJNS-10-00025-t007:** Expanded NBS parameters for the PNSP since the introduction of MS/MS.

Parameters	Value
Number of screened neonates	1,764,830
Global birth prevalence of IEM	1:2607
False positives (FPs)	2636
False negatives (FNs)	8
Positive predictive value (PPV)	21%
False positive rate (%)	0.15%
Sensitivity (%)	98.89%

## Data Availability

All data used in this research were obtained from the publications cited in this paper.

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
