# Peer review of "Portuguese Neonatal Screening Program: A Cohort Study of 18 Years Using MS/MS"

_2409-515X, 2024, doi:10.3390/ijns10010025_

Round 1
Reviewer 1 Report
Comments and Suggestions for Authors
- 2.2. Methods : the description of the sample preparation and choice of MS settings is missing (even it has probably evolved over the years, the description of the current method at least would be appreciated)
- Why don’t you choose SUAC in first tier for TYR 1? Did you never had a false negative tyrosinemia type I using Tyr as a first tier test during 18 years?
- Would it be possible to give more information about the 2636 results you got? For example, extending the table 6 (per year) to more diseases or groups of disease, giving a better overview of the diseases with a high rate of false positive (with or without 2TT)
- Do you have the data about the age at referral for a positive NBS test over the years?
- It might be surprising that in such a long period, you identified “only” 8 false negatives, did you specifically ask to all the reference centers if they did not found more cases of false negatives, diagnosed later in the life?
- You mention 4 cases of late-onset CPT2 deficiency, did you retrospectively assess the ratio C12/C0 at birth ? See also : https://www.mdpi.com/2409-515X/9/4/62
- In your paragraph 4, it might be interesting as well to discuss more about the potential additional diseases that could be screened in the future using MS/MS for NBS, potentially in a new table.
Author Response
Thank you very much for taking the time to review this manuscript.
Please see the attachment.

Reviewer 2 Report
Comments and Suggestions for Authors
This is an extensive compilation of data on metabolic disorders identified in the Portuguese Neonatal Screening Programme and is potentially very useful to neonatal screening programs and the metabolic clinics that assess neonates identified with positive screening findings.
It is interesting that 6 cases of tyrosinemia type I have been identified with elevated tyrosine in the program. This is quite different from the experience in Canada and the U.S. in that tyrosine has been a terrible marker for detecting TYRI by newborn screening (NBS) because almost all cases were missed when it was used as the primary screen. Consequently, programs in these two countries use SA for primary TYRI screening, not tyrosine. The difference seems to be the age of newborn specimen collection, in Portugal 3-6 days and allowing for an increase in tyrosine while in the U.S. and Canada specimen collection is 1-2 days. This merits a specific paragraph in Discussion with appropriate references. Included in the paragraph should be mention of any missed cases of TYRI or any of the tyrosinemias.
Similarly, 4 cases of homocystinuria (HCU) were identified using elevated methionine as the NBS marker while in the U.S. many cases have been missed because of a normal NBS methionine or one below the cutoff. Again, this difference is likely a function of the age of specimen collection. In fact, there can be a separate paragraph discussing the difference in ages of specimen collection and how that influences primary screening for both TYRI and HCU.
Author Response

(The authors gave the same response as above.)

Round 2
Reviewer 2 Report
Comments and Suggestions for Authors
Thank you for your responses.